# Adsorption Characteristics of Ag Nanoparticles on Cellulose Nanofibrils with Different Chemical Compositions

**DOI:** 10.3390/polym12010164

**Published:** 2020-01-08

**Authors:** Gu-Joong Kwon, Song-Yi Han, Chan-Woo Park, Ji-Soo Park, En-Ah Lee, Nam-Hun Kim, Madhusudhan Alle, Rajkumar Bandi, Seung-Hwan Lee

**Affiliations:** 1Kangwon Institute of Inclusion Technology, Kangwon National University, Chuncheon-Si, Gangwon-do 24341, Korea; gjkwon@kangwon.ac.kr; 2Department of Forest Biomaterials Engineering, College of Forest and Environmental Sciences, Kangwon National University, Chuncheon-si, Gangwon-do 24341, Korea; songlee618@kangwon.ac.kr (S.-Y.H.); chanwoo8973@kangwon.ac.kr (C.-W.P.); pojs04@kangwon.ac.kr (J.-S.P.); laa3158@kangwon.ac.kr (E.-A.L.); nhkim@kangwon.ac.kr (N.-H.K.); 3Institute of Forest Science, Kangwon National University, Chuncheon-si, Gangwon-do 24341, Korea; allemadhusudhan@gmail.com (M.A.); rajkumar.pgcb@gmail.com (R.B.)

**Keywords:** cellulose nanofiber, silver nanoparticles, adsorption, components, LCNF, HCNF, lignin

## Abstract

The adsorption characteristics of silver nanoparticles (AgNPs) on cellulose nanofibrils (CNFs) were investigated herein with different chemical compositions. Pure cellulose nanofibers (PCNFs), lignocellulose nanofibers (LCNFs) with different lignin contents (LCNF-20% and LCNF-31%), and holocellulose nanofibers (HCNFs) with hemicellulose were used in this study. Furthermore, CNFs and silver nitrate were mixed and reacted at different temperatures, and NaBH_4_ was used as the reducing agent. First, the effect of temperature on the adsorption of AgNPs on PCNF was studied. At an optimal temperature (45 °C), the effect of the chemical composition of CNF was studied. The overall properties were analyzed using UV-vis spectroscopy, transmission electron microscopy, X-ray diffraction, Fourier-transform infrared spectroscopy, and X-ray photoelectron spectroscopy. The AgNPs were found to be spherical under all conditions with average diameter of 5.3 nm (PCNF), 5.6 nm (HCNF), 6.3 nm (LCNF-20%) and 6.6 nm (LCNF-31%). The amount of AgNPs adsorbed on the CNF was observed to vary, based on the chemical composition of the CNF. The adsorption amount of AgNPs was observed to increase in the order of LCNF-20% > PCNF > LCNF-31% > HCNF. The results indicated that phenolic hydroxyl groups present in LCNF significantly affected the adsorption of AgNPs.

## 1. Introduction

Cellulose is highly abundant on earth, and is recognized as a renewable natural polymer that is chemically safe, environmentally sustainable, and possesses controllable properties [1,2,3]. It has long been used for daily life and industrial purposes, such as paper, cardboard, textiles, and building materials [4,5]. Recent research has focused on environmentally friendly functional new materials that use the physicochemical properties of cellulose, owing to the increase in the demand for renewable, biodegradable, and sustainable materials [6,7,8,9]. Among the forms of cellulose, more attention has been paid toward nano-sized cellulosic fibers, which have characteristics such as low weight, high strength, and high stiffness for preparing nanocomposites with inorganic and metallic particles [10,11]. Owing to the small size and high surface/volume ratio of the nanoparticles, nanocomposites are expected to exhibit new properties derived from synergistic optical, thermal, and mechanical properties [12,13,14].

Cellulose can best serve as a substrate for metal nanoparticles, owing to the electron-rich feature of the OH group found on its surface and its good colloidal stability in aqueous solutions [15,16]. Furthermore, the growth of metal nanoparticles can be controlled by the hydrogen bonding network formed by OH groups in the cellulose structure [17]. Cellulose can be used in various applications, such as for antibacterial activity, drug delivery, and biosensing [18,19]. Cellulose itself does not have antimicrobial properties; however, it can act as a supporting matrix for antimicrobial substances.

Silver nanoparticles (AgNPs) have received significant attention owing to their excellent physical and chemical properties as well as their high conductivity, excellent catalytic performance, and broad spectrum of antimicrobial activities, compared to other metals [20]. AgNPs can be synthesized using various methods, including electrochemical, gamma-radiation, photochemical, and ultrasound. The simplest known method is chemical reduction [13]. This method has been extensively studied, and many researchers have prepared cellulose/silver nanocomposites by optimizing variables such as the concentration of silver nitrate, cellulosic material, and other reaction conditions [21,22,23]. However, studies analyzing the effect of the chemical composition of the cellulosic material on the formation of AgNPs and its subsequent adsorption on cellulose nanofibers (CNFs) are relatively scarce.

On the other hand, various studies have reported that the chemical composition of raw materials, used in the production of cellulose nanofibrils, such as hemicellulose and lignin, significantly affects the defibrillation process. Previous studies have demonstrated that a high proportion of hemicellulose can improve the fibrillation process [24]. In addition, it has been reported that new nanomaterials with different chemical, mechanical, and surface characteristics compared to conventionally produced CNFs, can be obtained by including the lignin component [25]. This is expected to expand its application to newer fields that do not require the surface chemical functionalization of CNFs, and also to reduce the environmental stress and costs incurred during production [26]. Therefore, it is important to analyze the effect of the chemical composition of CNFs on the adsorption characteristics of AgNPs to develop new CNF/AgNPs-based functional materials.

In this study, we prepared AgNPs/CNF composites using the well-established chemical reduction method of silver ions by NaBH_4_. The effect of the chemical composition of CNF, in terms of the lignin and hemicellulose content, on the formation of the AgNPs, was examined in detail in this study. The main aim of this study was to investigate the adsorption characteristics of AgNPs on CNF. First, we studied the effect of temperature for the adsorption of AgNPs on pure cellulose nanofibril (PCNF). Subsequently, at the estimated optimal temperature, we studied the effect of the chemical composition of CNF, in terms of the lignin and hemicellulose content, on the adsorption of the AgNPs. Novelty of the present work lies in the usage of CNFs with different chemical composition for AgNPs adsorption.

## 2. Materials and Methods

### 2.1. Materials

Based on the chemical composition, PCNF, holocellulose nanofibril (HCNF), and lignocellulose nanofibril (LCNF) were used in this study. PCNF (BiNFi-S Cellulose) was purchased from Sugino Machine (Tokyo, Japan). Pine wood (*Pinus densiflora* Sieb. et Zucc.) powder (40–80 mesh) for preparing LCNF and HCNF was obtained from the Research Forest of Kangwon National University. Silver nitrate (AgNO_3_, 99%) and sodium borohydride (NaBH_4_, 98.0%) were purchased from Sigma-Aldrich, St. Louis, MO, USA.

#### 2.1.1. Preparation of HCNF and LCNFs

Two types of LCNF were prepared from the pine wood. LCNF, with a hemicellulose content of 26% and lignin content of 31% (LCNF-31%), was prepared from raw wood powder, without any chemical pretreatment. LCNF, with a hemicellulose content of 22% and lignin content of 20% (LCNF-20%), was obtained by pretreatment, using the alkaline hydrogen peroxide. For alkali pretreatment, a wood powder suspension (400 mL) was prepared, with 0.4% sodium hydroxide solution (1.6 g) and 5% solid concentration of wood powder (20 g). The slurry formed was then stirred at 170 rpm for 1 h, with the temperature maintained at 60 °C with a water bath. Furthermore, the insoluble residue was separated from the filtrate via vacuum filtration and the solid was washed with distilled water. Later, the alkali-pretreated sample (20 g) was added to a 12% hydrogen peroxide solution (980 mL), which was subsequently treated at 80 °C for 5 h, after adjusting the pH of the suspension to 11.5 using 50% sodium hydroxide solution. The solids were then washed with distilled water and vacuum-filtered until the pH of the wash filtrate became neutral.

Holocellulose, with cellulose (63%) and hemicellulose (37%), was obtained using the Wise method [27]. In this method, wood powder (20 g) and distilled water (1200 mL) were poured into a 2 L round flask and kept in a water bath at 80 °C for 20 min while being stirred at 150 rpm. The delignification reaction was initiated by adding sodium chlorite (8 g) and acetic acid (1600 µL) into the suspension and was continuously stirred for 1 h. Identical amounts of sodium chlorite and acetic acid were added every hour, and the process was repeated six times. Furthermore, the residue obtained was washed in a vacuum filter with distilled water until a neutral pH was reached. Subsequently, defibrillation was conducted using a high-pressure homogenizer (MN400BF; PICOMAX, Seoul, Korea) for producing LCNFs and HCNFs after pretreatment. The numbers mentioned after the abbreviation indicate the meaning of the reaction temperature.

#### 2.1.2. Chemical Composition

The lignin content was determined using the Klason lignin method. In this method, wood powder (1 g) was added to a 72% sulfuric acid solution (20 mL) at 20 °C for 2 h, while being gently stirred. Distilled water (345 mL) was subsequently added to a mixture containing dilute sulfuric acid at 3% concentration. Later, the diluted solution was placed in an autoclave at 121 °C for 1 h. Thereafter, the acid-insoluble residue was separated from the supernatant using a 1G4 glass filter. The cellulose and hemicellulose contents were determined from holocellulose using the following method. Holocellulose (30 g) was poured into a 17.5% sodium hydroxide solution (750 mL), and the reaction was performed for 50 min, under stirring at 150 rpm at a temperature between 20 and 23 °C. After 50 min, 10% acetic acid (750 mL) was added to the solution for neutralization. Later, the reactant was vacuum-filtrated and washed several times with distilled water. Subsequently, the two component contents were calculated by subtracting the obtained weight of cellulose from the weight of holocellulose.

### 2.2. Preparation of AgNPs and Adsorption on CNF

CNF-AgNPs were synthesized by modifying a method described earlier [28]. A 10 mM AgNO_3_ solution (500 μL) was added to 1% PCNF suspension (5 mL) and the mixture was stirred at 25, 45, 60, and 90 °C for 30 min at each temperature at 400 rpm. Thereafter, 20 mM NaBH_4_ solution (5 mL) was added to reduce the remaining AgNPs and was kept in dark surroundings for 24 h. This was followed by filtering and washing with distilled water, then drying for further analysis. Similarly, AgNPs-adsorbed LCNF-31%, LCNF-20%, and HCNF were prepared under optimal PCNF conditions (i.e., 45 °C).

### 2.3. Characterization

Preliminary characterization of the AgNPs was carried out using UV-visible spectroscopy. AgNPs reduced by the chemical method were monitored by measuring the UV-visible spectra of the solutions, after diluting the sample with deionized water five times. The spectra of the AgNPs solution were monitored by a UV-vis spectrophotometer (Libra s80, Biochrom Ltd., Cambridge, UK), within the wavelength range of 300 to 700 nm. Deionized water was used as a blank to adjust the baseline.

A field emission transmission electron microscope (JEM-2100F, JEOL, Tokyo, Japan), operating at 200 kV, was used to study the morphology and size of the AgNPs adsorbed on the CNFs. For the transmission electron microscopy (TEM) observation, a small droplet of diluted CNF/AgNPs suspension was placed on a carbon-coated copper grid. After washing, the CNF/AgNPs samples were negatively stained with a drop of 2% (w/v) uranyl acetate for 3 min and later dried at room temperature. The size of the AgNPs was reported as a diameter measured from 50 different AgNPs using Image J software (Version 1.52, Windows, free software, National Institute of Mental Health, Bethesda, MD, USA).

The crystal structure of the AgNPs was measured using an X-ray diffractometer (DMAX 2100 V, Rigaku Corp., Tokyo, Japan). The scanning range of 2*θ* was stepwise changed from 10° to 80°, and the scanning speed was maintained at 1°/min using the reflection method of Cu Kα radiation (λ = 1.5418 Å) at 40 kV and 30 mA.

Fourier transform infrared (FTIR) spectra of the CNF and AgNPs samples were analyzed using FTIR spectroscopy (Frontier, PerkinElmer, Waltham, MA, USA). FTIR analysis was performed using the attenuated total reflection method, and the spectra were recorded in the range of 4000–800 cm^−1^. Various modes of vibrations were identified and assigned to determine the different functional groups present in the samples.

The surface chemical changes of the CNF upon binding with AgNPs were examined using X-ray photoelectron spectroscopy (XPS, K Alpha +, Thermo Fisher Scientific, Waltham, MA, USA). Samples were irradiated by a monochromatic Al-Ka X-ray source (1486.6 eV). The spectra were acquired under a constant analyzer energy mode, with a pass energy of 150 eV and step size of 1.0 eV. Narrow scans were collected with a pass energy of 50 eV and step size of 0.1 eV. The binding energy scale of the XPS profile was calibrated by measuring the C1s peak at 284.5 eV. Digital acquisition and data processing were carried out using Thermo Scientific Avantage software (Thermo Fisher Scientific, East Grinstead, UK).

## 3. Results and Discussion

### 3.1. UV-Visible Spectroscopic Analysis

Figure 1 shows the UV-vis spectra of the reduced AgNPs on CNFs with different chemical compositions at different temperatures. UV-vis spectroscopy is an important technique for confirming the formation and stability of AgNPs in aqueous solutions. The size and shape of the AgNPs determine the spectral location and width of the absorption band (surface plasmon resonance (SPR) band). The UV-vis absorption of AgNPs is mostly known to exhibit a maximum within the range of 400–440 nm, due to their characteristics [29,30]. In this study, the effect of the reaction temperature on the formation and adsorption of AgNPs on PCNF was first investigated. As shown in Figure 1a, the intensity of the SPR peak increased as the temperature increased from 25 to 45 °C, and decreased at temperatures higher than 60 °C, exhibiting a maximum at 45 °C. Using this result, the temperature of 45 °C was selected to synthesize AgNPs on other CNFs (i.e., LCNF and HCNF). Digital photograph showing the aqueous dispersions of AgNPs adsorbed on different CNFs is shown in Appendix A. In this study, the wavelength of the absorption peak of AgNPs on PCNF (406 nm) did not change with the reaction temperature. However, peaks of AgNPs adsorbed on HCNF and LCNFs, under the same synthetic conditions, were observed at 412 and 416 nm, respectively (Figure 1b). This confirmed that the peak position for maximum absorption of the synthesized AgNPs changed, based on the chemical composition of the CNFs. Furthermore, the AgNPs absorbed light of different wavelengths based on their size and shape. The spherical form of AgNPs absorbed light within the wavelength range of 400–430 nm [11]. Thus, the absorption peaks within 406–416 nm suggested that the AgNPs obtained in this study were spherical. In summary, the wavelengths of the maximum absorption peaks increased in the order of PCNF < HCNF < LCNF. It is well-established that the wavelength of the SPR peak position is directly proportional to the particle size [11]. Hence, the increasing wavelength trend of the SPR peak, in the order of PCNF < HCNF < LCNF, implied that the sizes of AgNPs also increased in the same order. We assumed that PCNF, which had more hydroxyl groups, significantly controlled the growth of AgNPs during chemical reduction, whereas LCNF exhibited the least control. According to the study by James H. Johnston and Thomas Nilsson [31], lignin containing cellulosic fibers offers more binding sites (phenolic and methoxy groups attached to the electron rich aromatic rings of lignin) for the adsorption of nanosilver. However, the authors did not study the effect of lignin content. In our study, lignin containing CNFs, i.e., LCNF-20%, also exhibited more adsorption. However, increased lignin content (LCNF-31%) showed an adverse effect. Hence, we assume that the CNFs with moderate amounts of lignin can serve as good candidates for the adsorption of AgNPs.

### 3.2. TEM Analysis

Figure 2 shows the TEM images of the AgNPs adsorbed on PCNF, HCNF, LCNF-20%, and LCNF-31% and the corresponding histogram of the particle size distribution, obtained by analyzing 50 particles from different TEM images. All AgNPs showed a spherical morphology and were found to be uniformly distributed on the surface of all CNFs, without any aggregation. These results are similar to the results from the UV-vis spectra. The diameters of the AgNPs adsorbed on PCNF were found to be within 3–9 nm, and the average value was calculated to be 5.3 nm near the center of the particle size distribution. Similarly, the diameters of AgNPs adsorbed on HCNF were found to be within 3–10 nm, with an average of 5.6 nm. However, the diameters of AgNPs adsorbed on LCNF-20% and LCNF-31% were observed to be within 4–13 and 3–14 nm, with average diameters of 6.3 and 6.6 nm, respectively, which is higher than those for PCNF and HCNF. This result suggested that the size of AgNPs increased as the lignin content of LCNF increased (AgNPs on carbohydrates were smaller than on lignocellulosic fibers). This difference in the particle sizes suggested that the chemical composition of nanofibers also significantly controlled the growth of nanoparticles.

### 3.3. X-Ray Diffraction (XRD) Analysis

Figure 3 shows the XRD patterns of AgNPs adsorbed on different CNFs. In all the samples, four Bragg reflections were observed at 2*θ* angles of approximately 38.06°, 44.20°, 64.36°, and 77.32°. These reflections were indexed to the (111), (200), (220), and (311) diffraction planes of the face-centered cubic crystal structure of metallic silver (Joint Committee on Powder Diffraction Standards, silver file no. 04-0783), which indicated the highly crystalline nature of the formed AgNPs on CNFs. Compared to the intense (111) reflection, the (200), (220), and (311) reflections were found to be weaker and broader. This indicated that the nanocrystals were predominantly oriented along the (111) plane.

### 3.4. FTIR Analysis

Figure 4 shows the FTIR spectra of the CNFs, with and without AgNPs adsorption. Peaks at 3340 and 2900 cm^−1^, due to the stretching and symmetrical stretching of -OH and C-H, respectively, were observed in all the samples. The peak at 896 cm^−1^ represented C-H deformation in the cellulose. The peak at 1201 cm^−1^ was characteristic of PCNF and HCNF, due to -OH in-plane bending in the cellulose. The peak at 1735 cm^−1^ was characteristic of HCNF and LCNF, due to the stretching vibrations of C=O in acetyl and carboxyl groups in the hemicellulose [32]. Peaks at 1270, 1510, and 1595 cm^−1^ were attributed to the bending vibrations of C-O and stretching of C=C in the lignin, which were characteristic of LCNF with lignin. The peak at 1201 cm^−1^ was characteristic of PCNF and HCNF, due to OH in-plane bending in the cellulose. Furthermore, peaks at 1030 and 1060 cm^−1^ were due to C-O stretching or valence vibration from lignin and cellulose [33]. As shown in the figure, after the adsorption of AgNPs, the intensities of the peaks corresponding to the O-H, C-O, and C-H groups decreased, and the peaks widened. This demonstrated the binding interaction of these functional groups with AgNPs. Kim et al. [34] reported that the chemical and physical bonding between cellulose and silver ions occurs by substituting the hydrogen of the hydroxyl group of cellulose with Ag. Ali et al. [35] also reported that AgNPs adsorbed on cellulose exhibit typical IR peaks of cellulose; however, the peak at 3331 cm^−1^ due to the stretching vibration of the hydroxyl group was widened by the adsorption of AgNPs.

### 3.5. XPS Analysis

The adsorption characteristics of AgNPs on CNF were studied in detail using XPS. First, the effect of temperature on the adsorption of AgNPs on the PCNF was investigated. The in situ chemical reduction of Ag^+^ ions using NaBH_4_, and the subsequent adsorption of the formed AgNPs on PCNF, were studied at four different temperatures, i.e., 25, 45, 60, and 90 °C. Figure 5 shows the XPS profiles of PCNFs with AgNPs, obtained at different temperatures. The XPS profile of PCNF exhibits two significant peaks at approximately 284 and 530 eV, corresponding to C1s and O1s, respectively. Additionally, it can be observed that along with the peaks corresponding to C1s and O1s, a new peak, corresponding to Ag3d, is present for all four types of samples. High-resolution spectra of the Ag3d peak are shown in Figure 5c. The peak for Ag3d is divided into two peaks at 367.78 and 373.78 eV, corresponding to the Ag3d_5/2_ and Ag3d_3/2_ components of elemental silver, respectively. Generally, metallic Ag exhibited two peaks at 374.1 (Ag3d_5/2_) and 368.1 eV (Ag3d_3/2_), with a gap of 6.0 eV. However, in this case, the peaks were observed to exhibit a negative shift of 0.32 eV, compared to that of metallic Ag, but maintaining the gap of 6.0 eV. This negative shift indicated the binding interaction of AgNPs with the electron-rich groups of PCNF. These results confirmed the successful formation and adsorption of metallic AgNPs on PCNF, at all temperatures. However, changes in the temperature were observed to vary the amount of AgNPs adsorbed. Figure 5f shows the atomic percentages of Ag adsorbed on PCNF at different temperatures. As the temperature increased from 25 to 45 °C, the amount of adsorbed Ag also increased. However, as the temperature increased to 60 and 90 °C, the amount of adsorbed Ag was observed to decrease [36]. These results indicated that the temperature of 45 °C was optimal for adsorbing AgNPs under the given conditions—similar to the results from the UV-visible spectra (Figure 1a).

The effect of the chemical composition of CNFs on the adsorption of AgNPs was investigated using XPS. As shown in Figure 6, the survey profiles of all CNFs with AgNPs exhibited three significant peaks, corresponding to C1s, O1s, and Ag3d. Furthermore, the high-resolution Ag3d spectra (Figure 6c) exhibited two peaks at 373.58–373.78 eV and 367.58–367.78 eV, corresponding to Ag3d_3/2_ and Ag3d_5/2_, respectively, which indicated the metallic nature of the adsorbed AgNPs. Additionally, compared to PCNF and LCNF, both the peaks of Ag3d_3/2_ and Ag3d_5/2_ in HCNF exhibited a shift of 0.2 eV toward the lower binding energy (367.58 and 373.58 eV). This change can be attributed to the difference in the interaction between the adsorbed AgNPs and supporting substrate [37]. The atomic percentage of the adsorbed AgNPs is summarized in Figure 6f. LCNF-20% and HCNF exhibited the highest and lowest adsorptions of AgNPs, suggesting that the change in the chemical composition due to pretreatment affected the adsorption of the AgNPs. Furthermore, Li et al. [38] indicated that the amount of the phenolic hydroxyl group can be increased by alkaline hydrogen peroxide treatment. This indicated that the higher adsorption of AgNPs on LCNF-20% could be because of the increase in phenolic hydroxyl groups due to alkaline hydrogen peroxide treatment, and the lowest adsorption for HCNF could be due to excessive sodium chlorite-acetic acid treatment, compared to LCNF-31% from the raw wood without pretreatment. Lin et al. [39] investigated the capability of lignin and hemicellulose as reducing agents for synthesizing gold, platinum, and palladium nanoparticles. Both were found to exhibit good reactivity for reducing these metals; however, the metal particles were observed to be reduced by the hemicelluloses aggregated toward larger particles. Pang et al. [40] have reported that hemicellulose itself could not convert Ag^+^ to Ag^0^ and the formation rate of the nanoparticles increased with the addition of glucose. Oxygen atoms of the hydroxyl and carboxylate groups in hemicellulose contain a lone pair of electrons, enabling them to bind with Ag+ ions, which is subsequently reduced by the hemiacetal groups of glucose. It was suggested that the lowest AgNPs adsorption on HCNF was due to the core-shell structure of HCNF, where cellulose was covered by hemicellulose.

## 4. Conclusions

In this study, the effect of reaction temperature for the adsorption of AgNPs on PCNF and the adsorption characteristics of AgNPs on CNF, with different lignin and hemicellulose contents, were examined.

Results indicated that in UV-vis spectra, the SPR peak position of AgNPs adsorbed on PCNF was not changed by the reaction temperature. However, as the chemical composition of the CNF changed, the SPR peak position shifted toward longer wavelengths in the order PCNF < HCNF < LCNF-20% < LCNF-31%. Furthermore, the TEM analysis confirmed that AgNPs were formed in spherical shapes under all conditions with an average diameter of 5.3 nm (PCNF), 5.6 nm (HCNF), 6.3 nm (LCNF-20%) and 6.6 nm (LCNF-31%). The particle sizes of AgNPs were found to increase as the lignin content of CNF increased.

XRD analysis indicated a cubic crystal structure of metallic silver in all the samples, indicating high crystallinity of the AgNPs adsorbed on the CNF. Additionally, FTIR analysis revealed significant binding interactions between the surface functional groups of the CNF containing AgNPs. Furthermore, in the XPS profiles, the amount of AgNPs adsorbed on PCNF was found to vary with temperature. The highest adsorption was observed at 45 °C, and the adsorption was observed to decrease as the temperature increased. Additionally, the composition of CNF was observed to significantly affect the amount of AgNPs adsorbed. The highest and lowest amounts of AgNPs were found to be adsorbed on LCNF-20% and HCNF, respectively.

It was concluded from these results that the chemical composition of CNF significantly affected the adsorption of AgNPs.

## Figures and Tables

**Figure 1 polymers-12-00164-f001:**
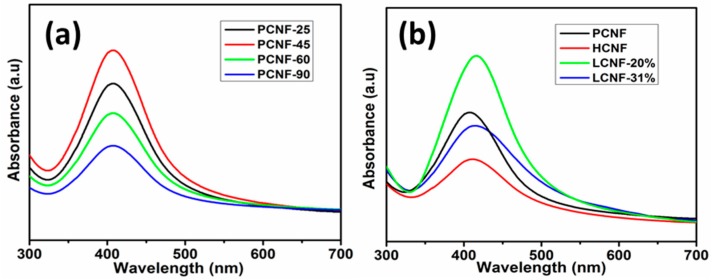
UV-visible absorption spectra of AgNPs adsorbed: (**a**) on PCNF at different temperatures; (**b**) on different CNFs with different chemical compositions at 45 °C.

**Figure 2 polymers-12-00164-f002:**
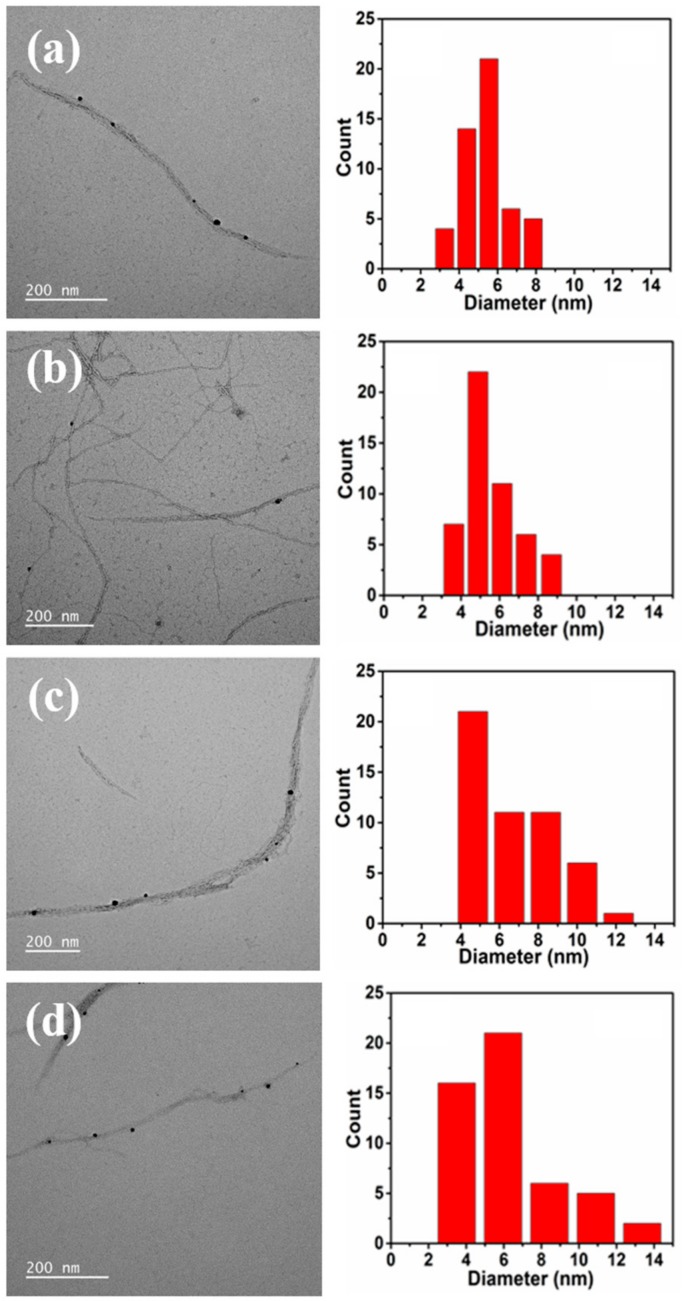
TEM images of AgNPs adsorbed on: (**a**) PCNF; (**b**) HCNF; (**c**) LCNF-20%; (**d**) LCNF-31%; and the corresponding histograms of the particle size distribution.

**Figure 3 polymers-12-00164-f003:**
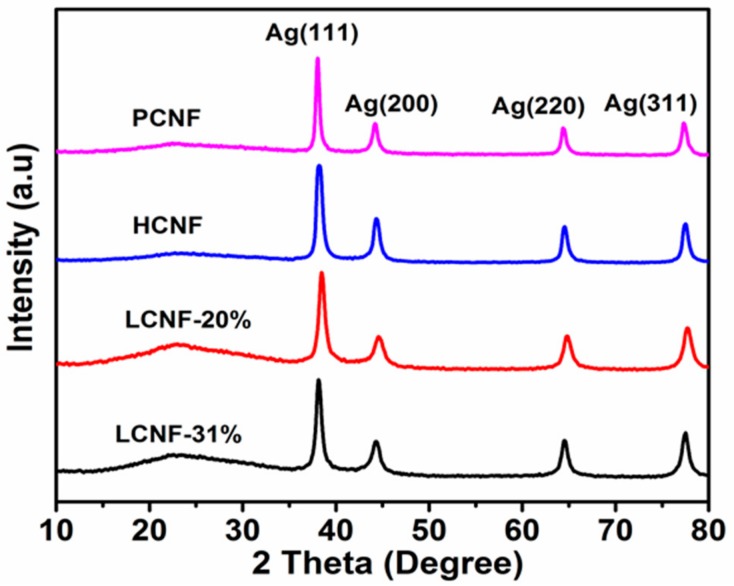
XRD patterns of AgNPs adsorbed on different CNFs at 45 °C.

**Figure 4 polymers-12-00164-f004:**
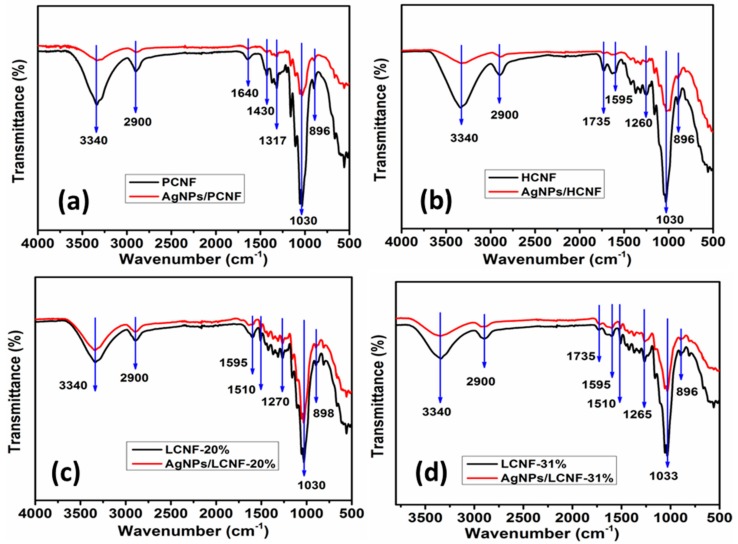
Comparison of FTIR spectra of the CNFs before and after the adsorption of AgNPs: (**a**) PCNF; (**b**) HCNF; (**c**) LCNF-20%; (**d**) LCNF-31% at the reaction temperature of 45 °C.

**Figure 5 polymers-12-00164-f005:**
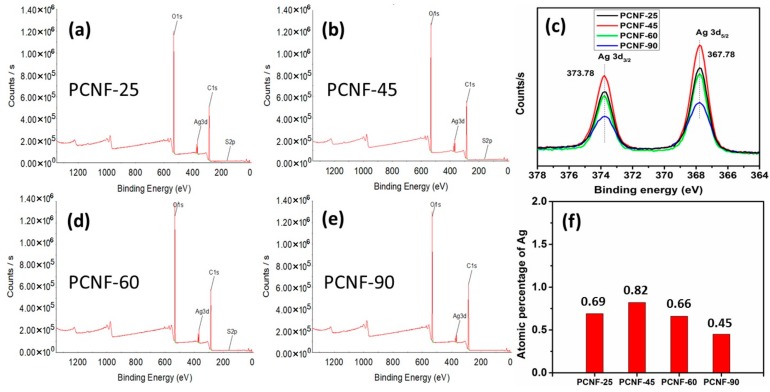
XPS survey profiles of AgNPs adsorbed on PCNF at different temperatures: (**a**) at 25 °C; (**b**) at 45 °C; (**d**) at 60 °C; and (**e**) at 90 °C. (**c**) High-resolution spectra of Ag3d; (**f**) atomic percentages of Ag in AgNPs adsorbed on PCNF at different temperatures.

**Figure 6 polymers-12-00164-f006:**
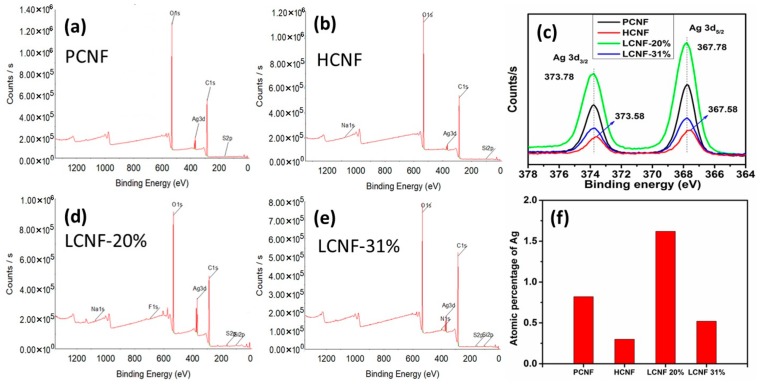
XPS survey profiles of AgNPs adsorbed on CNF with different contents, at 45 °C: (**a**) PCNF; (**b**) HCNF; (**d**) LCNF-20%; (**e**) LCNF-31%. (**c**) High-resolution Ag3d spectra of AgNPs adsorbed on CNF, with different contents, at 45 °C; (**f**) Atomic percentage of Ag in AgNPs adsorbed on CNF, with different contents, at 45 °C.

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
