# Peer review of "Adsorption Characteristics of Ag Nanoparticles on Cellulose Nanofibrils with Different Chemical Compositions"

_polymers, 2020, doi:10.3390/polym12010164_

Round 1
Reviewer 1 Report
Comments to the author: Major revisions after reconsideration
This work investigated the adsorption characteristics of silver nanoparticles (AgNPs) on cellulose nanofibrils with different chemical compositions. Even though the preparation of cellulose/silver nanocomposites is very common, it is still interesting to study the effect of the chemical composition of the cellulosic material on the formation of AgNPs. All experiments were systematically carried out and showed interesting results. However, there are still some issues, which do not meet the requirements of Polymers. Thus, I think the paper should be reconsidered with major revisions. Please see the comments below:
The major concern, the PCNF, HCNF and LCNF were obtained from two different companies. In other words, the chemical composition (cellulose, hemicellulose or lignin) and the raw materials were both different, this is less rigorous when designing experiments. Does the type of raw materials have effect on the formation of AgNPs?
Introduction needs more citations, such as the line 63-65 “In addition, it has been reported that new nanomaterials with different chemical, mechanical, and surface characteristics compared to conventionally produced CNFs, can be obtained by including the lignin component.” I give a few work that should be included:
(1) Rojo, E. et al. Comprehensive Elucidation of the Effect of Residual Lignin on the Physical, Barrier, Mechanical and Surface Properties of Nanocellulose Films. Green Chem., 2015, 17(3), 1853-1866.
(2) Bian, H. et al. Comparison of mixed enzymatic pretreatment and post-treatment for enhancing the cellulose nanofibrillation efficiency. Bioresource Technology, 2019, 122171.
(3) Bian, H. et al. Producing wood-based nanomaterials by rapid fractionation of wood at 80 °C using a recyclable acid hydrotrope. Green Chem., 2017, 19(14), 3370-3379.
Once these aforementioned points have been corrected this manuscript will make a good addition to the literature on LCNF production.
Line 182. It is very important to give a reasonable explanation on the effect of aromatic hydrophobic regions and phenolic groups of lignin on the absorption of AgNPs
Figure 1. Why the intensity of AgNPs adsorbed on the LCNF-31% decreased compared with LCNF-20%? More lignin on LCNF is not good for silver nanoparticle absorption?
Figure 4. Important peaks should be marked in the FTIR spectras.
I am very interested on the application of LCNF/silver nanocomposites, can authors list some specific work in the future?
Author Response
This work investigated the adsorption characteristics of silver nanoparticles (AgNPs) on cellulose nanofibrils with different chemical compositions. Even though the preparation of cellulose/silver nanocomposites is very common, it is still interesting to study the effect of the chemical composition of the cellulosic material on the formation of AgNPs. All experiments were systematically carried out and showed interesting results. However, there are still some issues, which do not meet the requirements of Polymers. Thus, I think the paper should be reconsidered with major revisions. Please see the comments below:
Comment 1: The major concern, the PCNF, HCNF and LCNF were obtained from two different companies. In other words, the chemical composition (cellulose, hemicellulose or lignin) and the raw materials were both different; this is less rigorous when designing experiments. Does the type of raw materials have effect on the formation of AgNPs?
Response 1: Only PCNF is obtained from a commercial source, other 3 types of CNFs were prepared from one source material i.e. pine wood powders. We used PCNF only as reference for comparison with other CNFs, hence we have selected the commercial one. (In view of the wide applicability of PCNF, we assume that comparing with the commercial grade sample will be more versatile).
Comment 2: Introduction needs more citations, such as the line 63-65 “In addition, it has been reported that new nanomaterials with different chemical, mechanical, and surface characteristics compared to conventionally produced CNFs, can be obtained by including the lignin component.” I give a few work that should be included:
(1) Rojo, E. et al. Comprehensive Elucidation of the Effect of Residual Lignin on the Physical, Barrier, Mechanical and Surface Properties of Nanocellulose Films. Green Chem., 2015, 17(3), 1853-1866.
(2) Bian, H. et al. Comparison of mixed enzymatic pretreatment and post-treatment for enhancing the cellulose nanofibrillation efficiency. Bioresource Technology, 2019, 122171.
(3) Bian, H. et al. Producing wood-based nanomaterials by rapid fractionation of wood at 80 °C using a recyclable acid hydrotrope. Green Chem., 2017, 19(14), 3370-3379.
Once these aforementioned points have been corrected this manuscript will make a good addition to the literature on LCNF production.
Response 2: Thank you for the suggestion. We have included these citations in the revised manuscript.
Comment 3: Line 182. It is very important to give a reasonable explanation on the effect of aromatic hydrophobic regions and phenolic groups of lignin on the absorption of AgNPs
Response 3: Thank you for the suggestion. Now we have included the explanation in the revised manuscript.
Comment 4: Figure 1. Why the intensity of AgNPs adsorbed on the LCNF-31% decreased compared with LCNF-20%? More lignin on LCNF is not good for silver nanoparticle absorption?
Response 4 : According to our observation, more lignin is not good for AgNPs adsorption (also evident from XPS analysis). Hence the intensity of AgNPs adsorbed on the LCNF-31% is decreased.
Comment 5: Figure 4. Important peaks should be marked in the FTIR spectras.
Response 5: Thank you for the suggestion, now we have marked the important peaks in the FTIR spectra (revised manuscript).
Comment 6: I am very interested on the application of LCNF/silver nanocomposites, can authors list some specific work in the future?
Response 6: Thank you for your interest. We are planning to study the antimicrobial or deordorizing applications of LCNF/silver nanocomposites in the future.
Reviewer 2 Report
Reviewers comments:
Kwang et al reported the adsorption characteristics of silver nanoparticles (AgNPs) on cellulose nanofibrils (CNFs) and were investigated herein with different chemical compositions. Work is well presented and interesting. Before its publication, some quarries should be responded.
The size of prepared CNCs and AgNPs should show in abstract and conclusion. Some recent papers by 2019 must insert in introduction. Nanomaterials 2019, 9 (11), 1523 and Carbohydrate polymers 2019,211, 181-194. The novelty of the work should mention at the end of introduction. The used AgNO3 and NaBH4 chemical details are missing. Check it. The dispersed nanoparticle in solvent digital image is needed. Line 254-259, regarding effect of temperature needs references. Check it.
Author Response
Kwon et al reported the adsorption characteristics of silver nanoparticles (AgNPs) on cellulose nanofibrils (CNFs) and were investigated herein with different chemical compositions. Work is well presented and interesting. Before its publication, some quarries should be responded.
Comment 1: The size of prepared CNCs and AgNPs should show in abstract and conclusion.
Response 1: Thank you for the suggestion. Changes were made accordingly.
Comment 2: Some recent papers by 2019 must insert in introduction. Nanomaterials 2019, 9 (11), 1523 and Carbohydrate polymers 2019,211, 181-194.
Response 2: suggested citations were included
Comment 3: The novelty of the work should mention at the end of introduction.
Response 3: Suggested changes were made
Comment 4: The used AgNO3 and NaBH4 chemical details are missing. Check it.
Response 4: We are sorry for the missing details. Now we have included them in the revised manuscript
Comment 5: The dispersed nanoparticle in solvent digital image is needed.
Response 5: Now we have provided a digital image as supplementary file.
Comment 6: Line 254-259, regarding effect of temperature needs references. Check it.
Response 6: thank you for the suggestion. We have added the reference in the revised manuscript
Round 2
Reviewer 1 Report
The author has well responded to the Reviewers' questions and revised the manuscript. It has reached the standard for publication in Polymers.
Reviewer 2 Report
Accepted.